# Comparative Analysis of Volatile Compounds and Biochemical Activity of *Curcuma xanthorrhiza* Roxb. Essential Oil Extracted from Distinct Shaded Plants

**DOI:** 10.3390/plants13192682

**Published:** 2024-09-25

**Authors:** Waras Nurcholis, Rahmadansah Rahmadansah, Puji Astuti, Bambang Pontjo Priosoeryanto, Rini Arianti, Endre Kristóf

**Affiliations:** 1Tropical Biopharmaca Research Center, IPB University, Bogor 16151, Indonesia; 2Department of Biochemistry, Faculty of Mathematics and Natural Sciences, IPB University, Bogor 16680, Indonesia; adanrahmadansah@apps.ipb.ac.id; 3Department of Biochemistry and Biomolecular Science, Faculty of Medicine, Universitas Tanjungpura, Pontianak 78124, Indonesia; pujiastuti@medical.untan.ac.id; 4Division of Veterinary Pathology, Faculty of Veterinary Medicine, IPB University, Bogor 16680, Indonesia; bpontjo@apps.ipb.ac.id; 5Department of Biochemistry and Molecular Biology, Faculty of Medicine, University of Debrecen, H-4032 Debrecen, Hungary; ariantirini@med.unideb.hu (R.A.); kristof.endre@med.unideb.hu (E.K.); 6Universitas Muhammadiyah Bangka Belitung, Pangkalpinang 33684, Indonesia

**Keywords:** *Curcuma xanthorrhiza* Roxb., essential oil, antioxidant, antibacterial, anticancer

## Abstract

The application of shade during plants’ growth significantly alters the biochemical compounds of the essential oil (EO). We aimed to analyze the effect of shade on the volatile compounds and biochemical activities of EO extracted from *Curcuma xanthorrhiza* Roxb. (*C. xanthorrhiza*) plants. Four shading conditions were applied: no shading (S0), 25% (S25), 50% (S50), and 75% shade (S75). The volatile compounds of EO extracted from each shaded plant were analyzed by gas chromatography–mass spectrometry. The antioxidant, antibacterial, and antiproliferative activities of EO were also investigated. We found that shade application significantly reduced the *C. xanthorrhiza* EO yield but increased its aroma and bioactive compound concentration. α-curcumene, xanthorrhizol, α-cedrene, epicurzerenone, and germacrone were found in EO extracted from all conditions. However, β-bisabolol, curzerene, curcuphenol, and γ-himachalene were only detected in the EO of S75 plants. The EO of the shaded plants also showed higher antioxidant activity as compared to unshaded ones. In addition, the EO extracted from S75 exerted higher antiproliferative activity on HeLa cells as compared to S0. The EO extracted from S0 and S25 showed higher antibacterial activity against Gram-positive bacteria than kanamycin. Our results suggest that shade applications alter the composition of the extractable volatile compounds in *C. xanthorrhiza,* which may result in beneficial changes in the biochemical activity of the EO.

## 1. Introduction

In their natural environment, plants typically grow in clusters, leading to inevitable mutual shading. These shading environments directly impact the light spectrum received by plants, typically characterized by the dominance of blue light (B), ranging between 400 and 500 nm and red (R) and far-red (FR) light between 600 and 700 nm and 700 and 750 nm, respectively. Additionally, there is a relatively low presence of ultraviolet-B light (UV-B) between 280 and 320 nm [1,2]. These light wavelengths are crucial for photosynthesis rates, thereby affecting plant growth and yield production [3,4,5,6]. The provision of shade, either natural or artificial, affects the amount and type of light received by plants through tailored shading, making this strategy valuable to mitigating global warming issues in crop production [7,8].

Several studies have demonstrated that shading significantly impacts both the production and composition of essential oil (EO) in various plants [9,10,11,12]. For instance, EO extracted from *Ocimum basilicum* L. cv. ‘Genovese’ under 50% blue net shading exhibited enhanced antimicrobial efficacy against *Staphylococcus aureus* (*S. aureus*), *Escherichia coli* (*E. coli*), and *Proteus vulgaris* (*P. vulgaris*) [12]. Furthermore, employing a 40% shade index for *Anethum graveolens* L. leads to increased EO yield and elevated activity against *Klebsiella pneumoniae* (*K. pneumoniae*) [13]. Understanding the influence of environmental factors on the production of secondary metabolites in medicinal plants is crucial due to their medical importance.

*Curcuma xanthorrhiza* Roxb. (*C. xanthorrhiza*), commonly known as Temulawak or Java turmeric, is a native herb from Indonesia with significant importance in traditional and herbal practices [14,15]. It plays a vital role in promoting the well-being of the Indonesian population through its various health benefits, including its antimicrobial and anticancer properties [16,17,18,19,20,21]. These therapeutic effects are attributed to the rich assortment of bioactive compounds found in its rhizomes, including terpenoids, phenols, flavonoids, saponins, alkaloids, and coumarins [14]. The rhizome, which serves as the source of active secondary metabolites such as curcuminoids and EOs, plays a crucial role in *C. xanthorrhiza* utilization and application for therapeutic purposes.

EOs comprise volatile and non-volatile compounds, which are secondary metabolites produced by various parts of plants [22]. Their production is influenced both by biotic and abiotic factors, including exposure to sunlight [23]. Modifying the light conditions through shading was shown to increase the percentage of EO production in thyme, marjoram, and oregano [24]. Moreover, shading may either enhance or diminish the production of specific secondary metabolites. For instance, the application of red-colored nets resulted in the highest percentage of patchoulol in *Pogostemon cablin* [11]. Similarly, employing 40% shading in *Anethum graveolens* L. enhanced the production of carvone but reduced the amount of limonene [13]. Nevertheless, the impact of shading on the EO composition of *C. xanthorrhiza* remains relatively unexplored to date.

Gas chromatography–mass spectrometry (GC-MS) has become a standard method in elucidating the components of EOs [25] and identifying metabolites in plant extracts [26,27,28]. In this study, we aimed to explore the chemical composition of *C. xanthorrhiza* EO in response to varying degrees of shading (0, 25, 50, and 75%) using GC-MS analysis. Chemometric analysis was employed to classify the shading levels of *C. xanthorrhiza*. Our investigation not only revealed the amount of the distinct volatile compounds but also evaluated the antioxidant, antibacterial, and antiproliferative properties of the EOs extracted from distinct shading conditions. This comprehensive examination aims to bridge gaps in our understanding of the EO composition of *C. xanthorrhiza* under different shading conditions, offering valuable insights into its potential as an antioxidant, antibacterial, and antiproliferative extract.

## 2. Materials and Methods

### 2.1. Chemicals

Ethanol (pro analysis), distilled water, HCl, BaCl_2_, H_2_SO_4_, ammonium acetate buffer, and trolox were purchased from Merck-Millipore (Darmstadt, Germany). 2,2-diphenyl-1-picrylhydrazyl (DPPH) powder was obtained from Himedia (Maharashtra, India). Ferric chloride (FeCl_3_) and 2,4,6-tripyridyl-s-triazine (TPTZ) were purchased from Sisco Research Laboratories Pvt. Ltd. (Maharashtra, India). *S. aureus*, *E. coli*, nutrient broth (NB), disc paper, kanamycin sulfate (KS), and muller hiton agar (MHA) were purchased from Sigma-Aldrich (St. Louis, MO, USA).

### 2.2. Sample Preparation and Shading Conditions

*C. xanthorrhiza* Roxb. varietas Cursina-III rhizomes with buds weighing 50–70 g were planted between January and September 2022 on open land within the Tropical Biopharmaca Cultivation Conservation, Unit Park of IPB University, located in Bogor, Indonesia, at an elevation of around 240 m above sea level, with geographic coordinates of 6°18′ 6°47′10 S and 106°23′45–107°13′30 W. This area has a wet tropical climate with rainfall with a range of temperature of 20–30 °C. All *C. xanthorrhiza* plants received the same conditions, including the type and amount of fertilizer, which contained urea at 250 kg/ha, SP-36, KCl each at 200 kg/ha, and 20 tons/ha of cow manure.

The plants were randomly grouped into 4 shade treatments: S0 (0%), S25 (25%), S50 (50%), and S75 (75%), with three replications for each group. The light grey-colored shades (AgroPro, PT. Gani Arta Dwitunggal, Padalarang, Indonesia) were composed of polyethylene-containing materials and configured as woven plastic nets. The density of the net was 66/127 holes/cm^2^. The percentage of shading represented the net density, which influenced the intensity of sunlight received by the plant. After 9 months, the rhizomes were harvested and immediately used for further analysis. Before processing, the plants were identified by an expert from the Tropical Biopharmaca Research Center, IPB University. Three kilograms of *C. xanthorrhiza* rhizomes from each shade condition was washed and chopped into 1–2 cm thickness for EO extraction, which was performed by using a simple steam distillation method. The samples were placed in distilled water for 4 h at 100–105 °C [28], and the yielded EO was used for further analysis.

### 2.3. Chemical Composition Analysis by Gas Chromatography–Mass Spectrometry (GC-MS)

The chemical composition of *C. xanthorrhiza* EO was analyzed by using Agilent Technologies 5977 GC-MS (Santa Clara, CA, USA). Three replicates of EO extraction were pooled and subjected to chemical compound analysis. The instrument was equipped with HP-5MS 5% PhenylMethylSilox (internal diameter 30 m × 250 µm with film thickness 0.25 µm) for the separation of the components. The GC oven temperature was initiated at 40 °C for 1 min, then gradually increased to 325 °C at a rate of 10 °C/min, and kept constant at 325 °C for 4 min. The relative amount of each identified volatile compound was quantified from the relative peak area of the chromatogram. Further confirmation of the chemical components was performed by comparing the retention time (RT) with the Willey 9 library database and PubChem data.

### 2.4. Antioxidant Activity Measurement

The following assays were carried out based on previously described methods [29] with modifications. Trolox was used as the standard, and the results were expressed as µmol of trolox equivalent (TE) per gram of dry weight (DW) or TE antioxidant capacity. DPPH and ferrous-reducing antioxidant activity (FRAP) assays were performed by using a modified nano-spectrophotometry method [30].

#### 2.4.1. DPPH Assay

Briefly, 100 µL of 500 µg/mL EO solution from each shade condition of *C. xanthorrhiza* was added to 100 µL of 125 M DPPH solution (in pro-ethanol analysis) in a 96-well microplate (Biologix Europe GmbH, Hallbergmoos, Germany). Next, the samples were homogenized and incubated for 30 min in the dark at room temperature. The absorbance of the EO solution from each shade condition of *C. xanthorrhiza* was measured using a SPECTROstar Nano (BMG LABTECH, Ortenberg, Germany) at 515 nm.

#### 2.4.2. FRAP Assay

Briefly, 10 µL of 1000 µg/mL *C. xanthorrhiza* EO from each shade condition and 300 µL of FRAP reagent (prepared by mixing acetate buffer at pH 3.6 with 10 M TPTZ solution (in 40 M HCl) and 20 µM FeCl_3_ (in distilled water) in a *v*/*v*/*v* ratio of 10:1:1) were placed in a 96-well microplate (Biologix Europe GmbH). The samples were then homogenized and incubated for 30 min in the dark at room temperature. The absorbance of the EO solution from each shade condition of *C. xanthorrhiza* was measured at a wavelength of 593 nm using SPECTROstar Nano (BMG LABTECH).

### 2.5. Antiproliferative Activity Analysis

All of the cell lines were purchased from American Type Culture Collection (ATCC, Manassas, VA, USA). Antiproliferative effects of the EO samples were measured on vero cells (CCL 81), a Michigan Cancer Foundation-7 (MCF-7) cell line for breast cancer (HTB 22), and a HeLa cell line for cervical cancer (CCL 2). Cells were seeded into 24-well plates at a density of 1 × 10^5^ cells/well. Cells were grown in DMEM/F-12 medium supplemented with 10% FBS and *C. xanthorrhiza* EO from each shade condition at a final concentration of 4 µg/mL. Doxorubicin (Sigma-Aldrich) 0.2 µg/mL was applied as a positive control. The cell cultures were incubated at 37 °C and 5% CO_2_ for 3 days. Next, the cells were harvested for cell counting by using the trypan blue method. A total of 90 µL of the cell suspension and 10 µL trypan blue solution was added into a 96-well plate and homogenized. The cell suspension and trypan blue solution (Sigma-Aldrich) were then dropped onto the end of the hemocytometer, which was covered with a cover glass until all parts under the cover glass were filled with the solution. The cell counting was performed using a light microscope (Optika, Ponteranica, Italy) at 40× magnification. Round cells located in 25 boxes at the center of the hemocytometer were counted. The total number of cells comprised the living and dead cells. The living cells were colorless, while the dead cells were blue. The number of cells per ml, percent proliferation, and antiproliferation rate were calculated using the following formula:$$\%\mathrm{growth}\mathrm{activity}=(\frac{\mathrm{Average}\mathrm{number}\mathrm{of}\mathrm{cells}\mathrm{by}\mathrm{treatment}}{\mathrm{Mean}\mathrm{number}\mathrm{of}\mathrm{negative}\mathrm{control}\mathrm{cells}})\times 100\%$$
% antiproliferative = 100% − (% growth activity)

### 2.6. Antibacterial Activity Analysis

Antibacterial activity analysis against *Staphylococcus aureus* (*S. aureus*) and *Escherichia coli* (*E. coli*) was performed by using the disk diffusion method. A total of 10 µL of bacterial culture was briefly rejuvenated in 30 mL of NB medium supplemented with *C. xanthorrhiza* EO (4 µg/mL) and incubated for 24 h at 37 °C. The bacterial culture was streaked onto the media (MHA) and incubated for 24 h. The clear zone around the disc was measured using a caliper to assess the antibacterial activity.

### 2.7. Statistical Analysis and Figure Preparation

All measured values are expressed as mean ± standard deviations (SDs) for the number of independent repetitions indicated. Statistical analysis was carried out using one-way ANOVA followed by Tukey’s post hoc test. GraphPad Prism 9 (GraphPad Software, San Diego, CA, USA) was used for figure preparation and statistical analysis.

## 3. Results

### 3.1. Shade Results in Strong Aroma, Concentrated C. xantorrhiza EO, and Decreased Extract Yield

*C. xanthorrhiza* EO was produced from mature and thick rhizomes by using water-steam distillation (hydrodistillation) (Figure 1a). We found that shade enhanced the aroma and concentration of the yielded *C. xanthorrhiza* EO in an intensity-dependent manner (Figure 1a). No shade (S0) and 25% shade (S25) yielded a light yellow EO with less aroma. On the other hand, 50% (S50) and 75% shade (S75) yielded a more concentrated *C. xanthorrhiza* EO with a dark brown color and stronger aroma. Shade treatment decreased the yield (% *v*/*w*) of EO in an intensity-dependent manner (Figure 1b). These data suggested that shade treatment on plants resulted in a more concentrated but lower yield of *C. xanthorrhiza* EO.

### 3.2. Sesquiterpenes Are the Major Chemical Compounds in C. xanthorrhiza EO

Next, we analyzed the chemical composition of *C. xanthorrhiza* EO by GC-MS. Identification of the chemical compounds is essential to predict the biological activity or potential medical benefits that can be provided by the plant extract. We detected 63 chemical compounds that belong to various groups, such as monoterpenes, carboxylic ester, dialkyl phosphate, diterpenoid, fatty alcohol, organosiloxane, triterpenes, and sesquiterpenes (Figure 2a). Our results showed that sesquiterpenes comprise 75% of total chemical compounds, which were detected by GC-MS, in *C. xanthorrhiza* EO extracted from four different shade conditions (Figure 2a and Table 1). α-curcumene, xanthorrhizol, α-cedrene, epicurzerenone, and germacrone were the chemical compounds with the highest amount (4–9%) and found in EO extracted from all shades (Figure 2b and Table 1). Several sesquiterpenes compounds, such as β-bisabolol, curzerene, curcuphenol, and γ-himachalene, were only detected in *C. xanthorrhiza* EO extracted from the S75 group (Figure 2b and Table 1). Our results showed that modifying the environmental condition by installing shade altered the abundance of chemical compounds, especially sesquiterpenes, which may contribute to the beneficial biological activities of *C. xanthorrhiza* EO.

### 3.3. Shade Enhanced the Antioxidant Activity of C. xanthorrhiza EO

Secondary metabolites can be attributed to the antioxidative activity of EOs. We observed that *C. xanthorrhiza* EO contains various active compounds and that S75 increased the abundance of several sesquiterpenes. Next, we analyzed the antioxidant activity of *C. xanthorrhiza* EO from each shade condition by using two different radical scavenging methods, DPPH and FRAP. Our results showed that S25 and S75 increased the free radical scavenging activity of *C. xanthorrhiza* EO as compared to S0 measured by the DPPH assay (Figure 3, left panel). On the other hand, S75 showed increased antioxidant activity measured by the FRAP assay as compared to S0 (Figure 3, right panel). Our results suggested that the shade enhanced the antioxidant activity of *C. xanthorrhiza* EO, which may provide benefits as a pharmacological agent.

### 3.4. Condition of 75% Shade Elevates the Antiproliferative Activity of C. xanthorrhiza EO in HeLa Cells

Active compounds, such as sesquiterpenes, are potential pharmaceutical agents for the treatment of cardiovascular diseases and cancers. First, we checked the antiproliferative activity of the extracts in non-cancerous vero cells, which originated from kidney epithelial cells of an African green monkey. We found that the EO extracted from shaded plants showed a lower antiproliferative activity on vero cells as compared to doxorubicin (Figure 4, left panel). However, no difference was observed between each shade condition (Figure 4, left panel). Next, we investigated the antiproliferative activity of *C. xanthorrhiza* EO from each shade condition by calculating the proliferation rates of the cancer cell lines MCF-7 and HeLa (Figure 4, middle and right panels, respectively). Our results showed that EOs from all shades exerted a comparable antiproliferative activity in vitro in both MCF-7 and HeLa cells as compared to doxorubicin (Figure 4). *C. xanthorrhiza* EO from S75 showed higher antiproliferative activity as compared to S0 in HeLa cells (Figure 4, right panel). A higher abundance of sesquiterpenes in S75 may contribute to the more effective antiproliferative activity, especially in HeLa cells.

### 3.5. C. xanthorrhiza EO Exhibits Antibacterial Activity

Next, we also evaluated antibacterial activity of *C. xanthorrhiza* EO by using the disk diffusion method. *S. aureus* (Gram-positive) and *E. coli* (Gram-negative) were used for the antibacterial evaluation, in which KS was administered as the positive control. Our results showed that *C. xanthorrhiza* EO exerted antibacterial activity at a comparable level as KS (Figure 5). The antibacterial test against *S. aureus* showed that the EO from S0 and S25 possessed higher antibacterial activity as compared to KS (Figure 5, left panel), while the EO from S50 and S75 had lower antibacterial activity as compared to S0 against *S. aureus* (Figure 5, left panel). On the other hand, *C. xanthorrhiza* EO from S50 showed lower antibacterial activity against *E. coli* as compared to KS (Figure 5, right panel). These results suggest that *C. xanthorrhiza* EO from S0 and S25 exerted a stronger antibacterial effect on Gram-positive bacteria as compared to KS but not on Gram-negative bacteria.

## 4. Discussion

*C. xanthorrhiza,* or ’temulawak’, is an indigenous Indonesian plant characterized by a pseudostem and typically reaches a height of 1.29–2.50 m [14,31]. In Indonesia, *C. xanthorrhiza* is often cultivated together with other plants, such as *Zea mays*, *Tectona grandis*, *Hevea brasiliensis*, *Azadirachta excelsa*, and *Albizia chinensis*, creating a shaded environment conducive to its growth [32,33,34,35,36]. Our study demonstrated a negative correlation between shading density and the yield of *C. xanthorrhiza* EO. A previous study with Chinese wild ginger (*Asarum heterotropoides* Fr. Schmidt var. mandshuricum (Maxim) Kitag.) reported that unshaded plants, which received full sunlight exposure, had higher yielded extract, including EOs, as compared to unshaded ones [37].

*C. xanthorrhiza* is a phototrophic organism, relying on light-dependent photosynthesis for its metabolic process, including the production of primary and secondary metabolites. Photosynthetic efficiency is influenced by CO_2_ concentration, transpiration, photosynthetic rate, and CO_2_ intake through stomata (stomatal conductance) [38,39]. Our previous study reported that shading did not affect the quantity of *C. xanthorrhiza* plant tillers; however, 75% shading significantly decreased the rhizome biomass [40]. Shading has been found to downregulate the expression of NADPH.quinone oxireductase (NQO), a protein essential for cellular respiration and CO_2_ intake, thereby reducing the photosynthetic capacity and stomatal index [41,42]. The highest yield of EO production in our S0 treatment likely occurred due to full light exposure, which elevated the stomatal index and subsequently increased the rate of photosynthesis [43]. In line with our results, a reduction in light intensity (25% shading) has also been shown to decrease the EO production of *Rosmarinus officinalis* L. by up to 43% [44].

Our study indicated that shading altered the secondary metabolites of *C. xanthorrhiza* EO. Terpenoid sesquiterpenes dominated the volatile compound identification; however, several sesquiterpenes were found only in EO extracted from 75% shaded plants. Reduced photosynthesis due to shading condition did not inhibit the secondary metabolite production. A previous study reported that the production and accumulation of volatile compounds are affected by the developmental stage of the plants [45,46]. In our study, the rhizomes were harvested from matured plants after 9 months of planting; therefore, the bioactive compounds could be produced and accumulated in the rhizomes during the plant’s growth [47]. Different spectra and intensities of light synergistically influence the chemical composition of many plants [9,13,24,48,49,50]. The variation of secondary metabolite content in *C. xanthorrhiza* EO extracted from different shade conditions resulted in different antioxidant activities measured in this study. The 75% shading condition significantly elevated the antioxidant activity assessed by both DPPH and FRAP analyses. The condition with 25% shade showed elevated antioxidant activity only measured by the DPPH test, which is more suitable for hydrophobic samples, such as EOs, whereas FRAP is more optimal for hydrophilic samples [51]. Therefore, the DPPH test was more sensitive to the antioxidant activity of EOs. Although 75% shading produced the lowest EO yield, the variety of its secondary metabolite content was in a wider range, in addition to its stronger color and aroma.

In addition to light, other external factors such as temperature, CO_2_ concentration, and moisture can also affect the composition of both primary and secondary metabolites of plants [52]. The response of plants to climate change may alter the active compound composition that is attributed to the pharmaceutical properties of the plants. The plants from which EOs were extracted were planted at temperatures ranging from 20 to 30 °C, which support healthy growth without exposing the plants to cold or heat stress. Increased CO_2_ levels can also affect the plants’ growth and development and elevate biomass production. Plants growing at elevated CO_2_ levels typically exhibit reduced nutritional quality and nitrogen and protein content due to enhanced dilution of nutrients [52]. However, the response can be different depending on the duration of the exposure and other environmental factors. Another study conducted using *C. xanthorrhiza* (WP.5818) obtained from the market reported that the volatile compounds of the extracted EO were dominated by monoterpenes [53]. The difference in the environmental factors may cause this discrepancy with our study, which showed that sesquiterpenes built 75% of the volatile compounds.

Numerous studies have shown a positive correlation between antioxidant activity and antiproliferative effects against various cancers [54,55,56,57,58]. In association with the currently reported data, our previous study found that *C. xanthorrhiza* EO did not affect the proliferation of vero cells irrespective of the concentration, indicating the selectivity of the EO [59]. We also examined the effect of the EO samples on two cancer cell lines: MCF-7 for breast cancer and HeLa for cervical cancer. The 75% shading condition resulted in higher antiproliferative activity of the EO in HeLa cells compared to the unshaded condition. In our study, 75% shading stimulated the production of more chemical compounds, such as β-bisabolol, curzerene, curcuphenol, and γ-himachalene, which have been reported to possess anticancer properties in several studies [60,61,62,63,64]. Curzerene (C_15_H_20_O), a sesquiterpene commonly found in Curcuma rhizomes, was shown to suppress the proliferation of glioblastoma, melanoma, and human lung adenocarcinoma cells [62,63,65]. Curzerene was found to downregulate the expression of glutathione S-transferase (GST) A1 and A4 at the mRNA level, leading to apoptosis by preventing DNA damage during replication in the G0/G1 phase [62,63]. Additionally, curzerene inhibited cell proliferation by suppressing the activation of the mammalian target of the rapamycin (mTOR) pathway [62].

Bisabolol, which consists of α-bisabolol and β-bisabolol, is an active monocyclic sesquiterpene alcohol known for its anti-inflammatory and skin-soothing effects [66]. Research on β-bisabolol is less extensive than on its isomer, α-bisabolol. Several studies reported that β-bisabolol has similar anti-inflammatory properties, suggesting comparable pharmacological potential [67]. The α-bisabolol exhibits antiproliferative and cytotoxic effects, leading to apoptosis of various cancer cell lines [68,69,70,71]. These apoptotic effects are stimulated via the inhibition of the transducer and activator of transcription 3 (STAT-3), a transcription factor crucial in tumorigenesis and cancer progression [72]. Similarly, β-bisabolol from *Semenovia suffruticosa* has been demonstrated to exert cytotoxic properties against several types of cancer cells [73]. Curcuphenol and γ-himachalene, compounds found in *C. xanthorrhiza* grown under 75% shading, are also associated with apoptotic mechanisms in cancers. Curcuphenol induces apoptosis by enhancing the expression of caspase-3 and antigen processing and presentation machinery (APM), leading to a stronger immune response against metastatic tumor cells [64,74,75]. In addition, the anticancer properties of γ-himachalene may result from its cytotoxic effects [76,77,78].

Although sesquiterpenes, such as γ-himachalene and bisabolol, were identified in plants subjected to 75% shading, we did not find any noticeable change in the antibacterial properties of the corresponding EO [79,80,81,82]. Interestingly, our investigation found that the antibacterial efficacy was pronounced in *C. xanthorrhiza* plants cultivated under full sun exposure and with only 25% shading, particularly against *S. aureus*, a Gram-positive bacterium. Through comprehensive GC-MS analysis, we observed that these treatments exhibited less complex secondary metabolic profiles compared to denser shading conditions (50% and 75%). These results suggested that the observed antibacterial effect may be attributed to the presence of α-curcumene and α-cedrene, compounds found in higher concentrations in plants grown under full sun exposure or with 25% shading as compared to those subjected to 50% or 75% shading. Various studies have demonstrated a positive correlation between α-cedrene concentrations and the inhibition of *S. aureus* growth [83,84]. Additionally, α-curcumene has been shown to be effective against Gram-positive bacteria [85,86,87] by disrupting their cell membrane, leading to bacterial cell death [88].

## 5. Conclusions

The study highlights the significant impact of shading on the yield, composition, and bioactive properties of *C. xanthorrhiza* Roxb. EO. We found that increased shading density negatively correlated with EO yield, and full sunlight exposure enhanced its production due to increased photosynthetic efficiency. Shading also influenced secondary metabolite profiles. The 75% shading condition significantly boosted the antioxidant activity despite resulting in the lowest yield. This condition produced a broader variety of secondary metabolites, contributing to stronger color and aroma. Furthermore, EO from plants under 75% shading showed higher antiproliferative activity against HeLa cancer cells, attributed to compounds like β-bisabolol, curzerene, curcuphenol, and γ-himachalene. Interestingly, while sesquiterpenes, such as γ-himachalene and bisabolol, were prevalent under 75% shading, they lacked notable antibacterial properties, whereas plants under full sun and 25% shading exhibited strong antibacterial effects against *S. aureus*, linked to higher concentrations of α-curcumene and α-cedrene. Our findings highlight the importance of optimal shading conditions in maximizing the therapeutic potential of *C. xanthorrhiza*, paving the way for its broader applications in the pharmaceutical and healthcare sectors.

## Figures and Tables

**Figure 1 plants-13-02682-f001:**
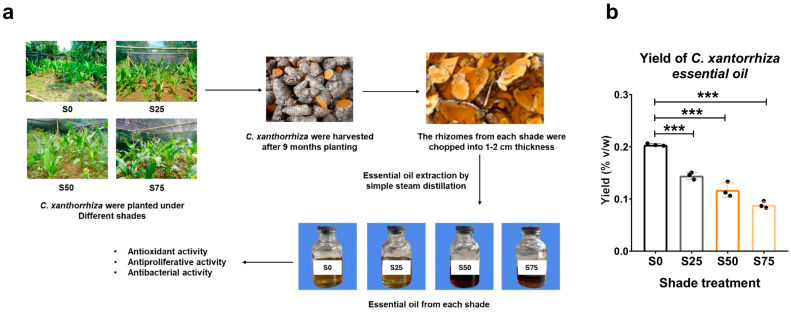
(**a**) The workflow of essential oil extraction from *C. xanthorrhiza* Roxb. rhizomes. (**b**) Yield (% *v*/*w*) of *C. xanthorrhiza* essential oil. Extraction was performed in 3 independent plants, with statistical analysis by one-way ANOVA followed by Tukey’s post hoc test, *** *p* < 0.001. S0, no shade; S25, 25% shade; S50, 50% shade; S75, 75% shade.

**Figure 2 plants-13-02682-f002:**
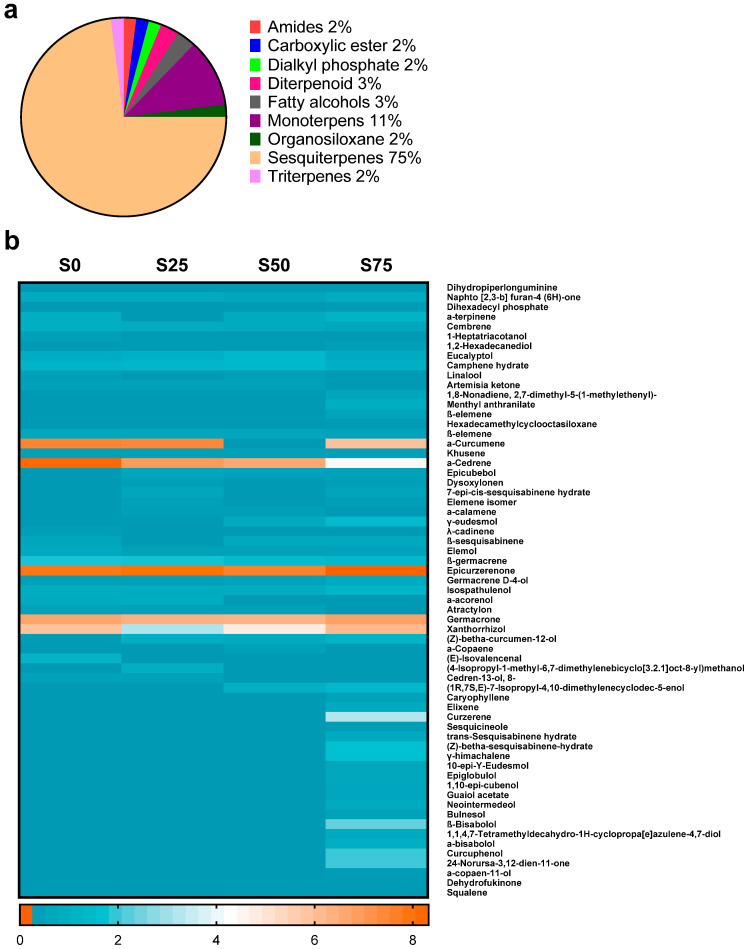
Chemical compositions of *C. xanthorrhiza* Roxb. essential oil. (**a**) Pie chart displaying the groups of secondary metabolites in *C. xanthorrhiza* essential oil from 4 shading conditions detected by GC-MS. (**b**) Heatmap displaying the abundance of 64 compounds detected by GC-MS in *C. xanthorrhiza* essential oil from each shade condition. Three replicates of EO extraction were pooled and subjected to chemical compound analysis.

**Figure 3 plants-13-02682-f003:**
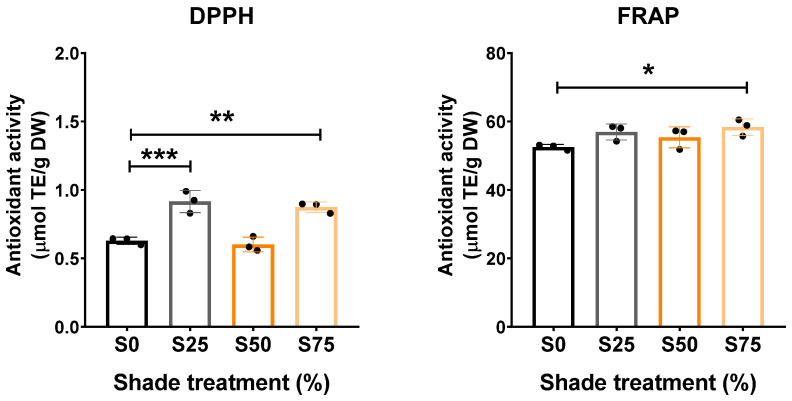
Antioxidant activity of *C. xanthorrhiza* Roxb. essential oil measured by DPPH and FRAP assays. n = 3, statistical analysis by one-way ANOVA followed by Tukey’s post hoc test, * *p* < 0.05, ** *p* < 0.01, *** *p* < 0.001. S0, no shade; S25, 25% shade; S50, 50% shade; S75, 75% shade. DPPH, 2,2′-diphenyl-1-picrylhydrazyl; DW, dry weight; FRAP, ferric reducing antioxidant power; TE, Trolox equivalent; DW: dry weight.

**Figure 4 plants-13-02682-f004:**
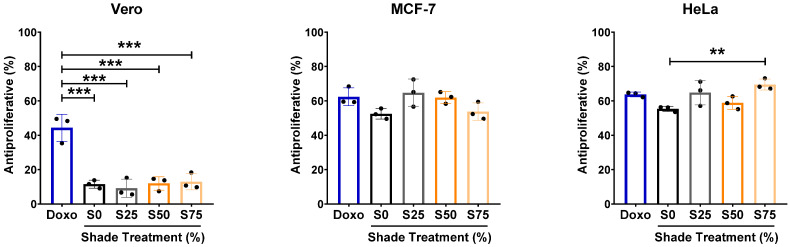
Antiproliferative activity of *C. xanthorrhiza* Roxb. essential oil (4 µg/mL) measured in vero (**left**), MCF-7 (**middle**), or HeLa (**right**) cells. n = 3, statistical analysis by one-way ANOVA followed by Tukey’s post hoc test, ** *p* < 0.01, *** *p* < 0.001. S0, no shade; S25, 25% shade; S50, 50% shade; S75, 75% shade. Doxo: doxorubicin (0.2 µg/mL).

**Figure 5 plants-13-02682-f005:**
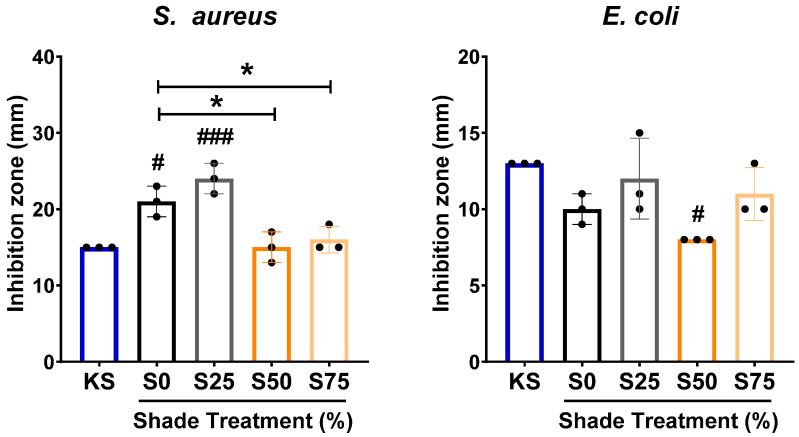
Antibacterial activity of *C. xanthorrhiza* Roxb. essential oil (4 µg/mL) tested against *S. aureus* (**left**) or *E. coli* (**right**). n = 3, statistical analysis by one-way ANOVA followed by Tukey’s post hoc test, */# *p* < 0.05, ### *p* < 0.001. * Analysis was performed by using S0 as reference. # Analysis was performed by using KS as reference. S0, no shade; S25, 25% shade; S50, 50% shade; S75, 75% shade. KS: kanamycin sulfate (1 mg/mL).

**Table 1 plants-13-02682-t001:** Volatile compounds identified from the essential oils of various shades of *C. xanthorrhiza.* Roxb. Three replicates of EO extraction were pooled and subjected to chemical compound analysis.

No.	Compounds	Group Compounds	MF	MW (g/mol)	RT				%Area			
					N0%	N25%	N50%	N75%	N0%	N25%	N50%	N75%
**1**	Eucalyptol	Monoterpenes	C_10_H_18_O	154	12.566	12.562	12.560	12.551	0.69	0.88	1.22	0.60
**2**	Camphene hydrate	Monoterpenes	C_10_H_18_O	154	16.714	16.706	16.700	16.689	0.99	1.18	1.19	0.79
**3**	β-elemene	Sesquiterpenes	C_15_H_24_	204	23.486	23.433	23.442	23.433	0.49	0.47	0.43	0.36
**4**	α-Curcumene	Sesquiterpenes	C_15_H_22_	202	25.918	25.775	-	25.767	7.40	7.29	-	5.81
**5**	Khusene	Sesquiterpenes	C_15_H_24_	204	26.317	26.308	26.308	26.300	0.26	0.27	0.28	0.23
**6**	α-Cedrene	Sesquiterpenes	C_15_H_24_	204	26.656	26.625	26.483	26.603	8.11	6.79	6.44	4.32
**7**	Epicubebol	Sesquiterpenes	C_15_H_26_O	222	-	26.745	26.667	26.736	-	0.41	0.36	0.28
**8**	Dys-oxylonen	Sesquiterpenes	C_15_H_24_	204	-	26.817	-	26.800	-	0.17	-	0.13
**9**	7-epi-cis-sesqui-sabinene hydrate	Sesquiterpenes	C_15_H_26_O	222	-	26.875	-	26.865	-	0.43	-	0.37
**10**	Elemene isomer	Sesquiterpenes	C_15_H_24_	204	-	26.982	-	26.974	-	0.27	-	0.18
**11**	α-calamene	Sesquiterpenes	C_15_H_22_	202	-	27.417	27.392		-	0.20	0.17	
**12**	γ-eudesmol	Sesquiterpenes	C_15_H_26_0	222	-	-	29.507	30.043	-	-	0.61	1.34
**13**	λ-cadinene	Sesquiterpenes	C_15_H_24_	204	26.833	-	-	-	0.16	-	-	-
**14**	β-sesqui-sabinene	Sesquiterpenes	C_15_H_24_	204	26.890	-	26.872	26.872	0.42	-	0.55	0.55
**15**	Elemol	Sesquiterpenes	C_15_H_26_O	222	27.488	27.479	27.476	27.481	0.55	0.31	0.29	0.35
**16**	Linalool	Monoterpenes	C_10_H_18_0	154	27.742	-	27.733	-	0.18	-	0.19	-
**17**	β-germacrene	Sesquiterpenes	C_15_H_24_	204	27.826	27.813	27.810	27.810	1.68	1.65	1.39	1.28
**18**	Epi-curzerenone	Sesquiterpenes	C_15_H_18_O_2_	230	28.821	28.813	28.800	28.854	7.72	7.89	7.45	8.32
**19**	α-terpinene	Diterpenoid	C_20_H_32_	272	28.971	-	28.956	28.978	0.74	-	0.59	0.94
**20**	Artemisia ketone	Monoterpenes	C_10_H_16_O	152	29.042	29.042	29.042	-	0.25	0.28	0.26	-
**21**	Germacrene D-4-ol	Sesquiterpenes	C_15_H_26_O	222	29.267	29.258	29.258	29.263	0.20	0.19	0.27	0.63
**22**	Iso-spathulenol	Sesquiterpenes	C_15_H_24_O	220	29.408	29.411	29.402	29.416	0.72	0.68	0.70	1.07
**23**	α-acorenol	Sesquiterpenes	C_15_H_26_0	222	29.557	29.553	-	-	0.58	0.73	-	-
**24**	1-Heptatria-cotanol	Fatty alcohols	C_37_H_76_O	536	29.632	-	-	-	0.20	-	-	-
**25**	Atractylon	Sesquiterpenes	C_15_H_20_O	216	30.158	30.158	30.150	-	0.28	0.30	0.25	-
**26**	Germacrone	Sesquiterpenes	C_15_H_22_0	218	31.000	30.988	30.978	31.015	6.54	6.24	6.13	6.62
**27**	Cembrene	Diterpenoid	C_20_H_32_	272	31.553	31.533	31.532	26.452	0.80	0.67	0.64	0.42
**28**	Xantorrhizol	Sesquiterpenes	C_15_H_22_O	218	32.306	32.289	32.288	32.283	5.80	3.05	4.66	6.03
**29**	Naphto [2,3-b] furan-4 (6H)-one	Carboxylic ester	C_15_H_18_0_2_	230	32.414	32.398	32.397	32.435	0.60	0.58	0.41	0.67
**30**	(Z)-betha-curcumen-12-ol	Sesquiterpenes	C_15_H_24_O	220	32.467	29.962	29.163	29.175	0.09	0.75	0.64	1.05
**31**	α-Copaene	Sesquiterpenes	C_15_H_26_O_2_	238	-	-	32.742	-	-	-	0.30	-
**32**	(E)-Iso-valencenal	Sesquiterpenes	C_15_H_22_O	218	32.628	-	-	-	0.95	-	-	-
**33**	(4-Isopropyl-1-methyl-6,7-dimethylenebicyclo [3.2.1]oct-8-yl)methanol	Sesquiterpenes	C_15_H_24_O	220	-	32.614	-	-	-	0.81	-	-
**34**	Cedren-13-ol, 8-	Sesquiterpenes	C_15_H_24_O	220	32.718	32.704	-	-	0.32	0.30	-	-
**35**	(1R,7S,E)-7-Isopropyl-4,10-dimethylenecyclodec-5-enol	Sesquiterpenes	C_15_H_24_O	220	-	-	32.160	32.644	-	-	0.91	1.21
**36**	Squalene	Triterpenes	C_30_H_50_	410	32.833	-	32.833	-	0.07	-	0.07	-
**37**	Caryo-phyllene	Sesquiterpenes	C_15_H_24_	204	-	-	-	24.299	-	-	-	0.31
**38**	Elixene	Sesquiterpenes	C_15_H_24_	204	-	-	-	24.525	-	-	-	0.62
**39**	Curzerene	Sesquiterpenes	C_15_H_20_O	216	-	-	-	26.125	-	-	-	3.07
**40**	Sesqui-cineole	Sesquiterpenes	C_15_H_26_O	222	-	-	-	26.676	-	-	-	0.11
**41**	trans-Sesqui-sabinene hydrate	Sesquiterpenes	C_15_H_26_O	222	-	-	-	27.351	-	-	-	0.63
**42**	(Z)-betha-sesqui-sabinene-hydrate	Sesquiterpenes	C_15_H_26_O	222	-	-	-	27.631	-	-	-	1.54
**43**	γ-himachalene	Sesquiterpenes	C_15_H_24_	204	-	-	-	28.501	-	-	-	1.56
**44**	10-epi-Y-Eudesmol	Sesquiterpenes	C_15_H_26_O	222	-	-	-	29.483	-	-	-	0.41
**45**	Epiglobulol	Sesquiterpenes	C_15_H_26_O	222	-	-	-	29.557	-	-	-	0.53
**46**	1,8-Nonadiene, 2,7-di-methyl-5-(1-methyl-ethenyl)-	Monoterpenoids	C_14_H_24_	192	-	-	-	28.632	-	-	-	0.42
**47**	1,10-epi-cubenol	Sesquiterpenes	C_15_H_26_O	222	-	-	-	29.750	-	-	-	0.44
**48**	Guaiol acetate	Sesquiterpenes	C_17_H_28_O_2_	264	-	-	-	29.833	-	-	-	0.44
**49**	Neo-intermedeol	Sesquiterpenes	C_15_H_26_O	222	-	-	-	30.117	-	-	-	0.60
**50**	Bulnesol	Sesquiterpenes	C_15_H_26_O	222	-	-	-	30.258	-	-	-	0.28
**51**	β-Bisabolol	Sesquiterpenes	C_15_H_26_O	222	-	-	-	30.368	-	-	-	2.18
**52**	1,1,4,7-Tetramethyldecahydro-1H-cyclopropa[e]azulene-4,7-diol	Sesquiterpenes	C_15_H_26_O_2_	238	-	-	-	30.531	-	-	-	0.61
**53**	α-bisabolol	Sesquiterpenes	C_15_H_26_O	222	-	-	-	30.739	-	-	-	0.79
**54**	Menthyl anthranilate	Monoterpenoids	C_17_H_25_NO_2_	275	-	-	-	30.833	-	-	-	0.76
**55**	Curcu-phenol	Sesquiterpenes	C_15_H_22_O	218	-	-	-	31.272	-	-	-	1.83
**56**	β-elemene	Monoterpenoids	C_14_H_24_	192	-	-	-	32.864	-	-	-	0.37
**57**	24-Norursa-3,12-dien-11-one	Sesquiterpenes	C_29_H_44_O	408	-	-	-	32.087	-	-	-	1.86
**58**	α-copaen-11-ol	Sesquiterpenes	C_15_H_24_O	220	-	-	-	33.202	-	-	-	0.10
**59**	Dihydro-piperlonguminine	Amides	C_16_H_21_NO_3_	275	-	-	-	33.983	-	-	-	0.07
**60**	Dehydro-fukinone	Sesquiterpenes	C_15_H_22_O	218	-	-	-	35.143	-	-	-	0.03
**61**	1,2-Hexadecane-diol	Fatty Alcohols	C_16_H_34_O_2_	258	-	-	-	36.334	-	-	-	0.17
**62**	Hexa-decamethyl-cyclo-octasiloxane	Organosiloxane	C_16_H_48_O_8_Si_8_	592	-	-	-	38.479	-	-	-	0.08
**63**	Dihexadecyl phosphate	Dialkyl phosphate	C_32_H_67_O_4_P	546	-	-	-	47.614	-	-	-	0.09

## Data Availability

The original contributions presented in the study are included in the article, and further inquiries can be directed to the corresponding author/s.

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
