# Peer review of "Comparative Analysis of Volatile Compounds and Biochemical Activity of *Curcuma xanthorrhiza* Roxb. Essential Oil Extracted from Distinct Shaded Plants"

_plants, 2024, doi:10.3390/plants13192682_

Round 1

Reviewer 1 Report (Previous Reviewer 2)

Comments and Suggestions for Authors

The authors corrected the article according to my suggestions. They added an analysis of the effects of compounds on mammalian cells. The article can be accepted. They should still only review the notation of Latin names because they made numerous mistakes in the literature.

Author Response

We would like to thank Reviewer 1 for the valuable comment. We have checked and corrected the Latin names in the revised version of the manuscript.

Reviewer 2 Report (New Reviewer)

Comments and Suggestions for Authors

The manuscript by Nurcholis et al. explored volatiles and biochemical activity of distinct shaded Curcuma xanthorrhiza essential oil. In my opinion, the manuscript can be accepted after minor revisions. Please see my comments to improve the manuscript.

-Although the authors studied and summarized the effect of shade on essential oil composition, in practical applications, the materials for extraction usually come from different batches and may have different origins and varieties. The authors need to elaborate on the practical implications of this study.

-A summary of other possible factors affecting the composition of essential oil, such as light, temperature, gases, etc., is missing. Mechanisms of changes in essential oil composition, such as the effects of plant secondary metabolism, also need to be addressed.

-Line 102-103, the author should provide the actual picture illustrating how to create different shade proportions.

-Line 104, how exactly was the density of the woven plastic nets?  The manufacture also need be provided.

-Line 165-170, how many volume of essential oil was dipped on the plates?

-There is a lack of data on essential oil production under different shade conditions, such as yield.

Author Response

The manuscript by Nurcholis et al. explored volatiles and biochemical activity of distinct shaded Curcuma xanthorrhiza essential oil. In my opinion, the manuscript can be accepted after minor revisions. Please see my comments to improve the manuscript.

Answer: We would like to thank Reviewer 2 for valuable comments and insight. We have revised the manuscript accordingly and point-by-point response can be seen below.

-Although the authors studied and summarized the effect of shade on essential oil composition, in practical applications, the materials for extraction usually come from different batches and may have different origins and varieties. The authors need to elaborate on the practical implications of this study.

Answer: In this study, we used Curcuma xanthorrhiza variety Cursina-III (this information has been added, line 97) from the same batch. The plants used in this study have been validated by the expert in Tropical Biopharmaca Cultivation Conservation, Bogor, Indonesia.

-A summary of other possible factors affecting the composition of essential oil, such as light, temperature, gases, etc., is missing. Mechanisms of changes in essential oil composition, such as the effects of plant secondary metabolism, also need to be addressed.

Answer: The information regarding the temperature and humidity condition in the planting site has been added (line 100-102). The discussion about other possible factors which may affect the composition of essential oil has been added as well (line 323-336).

-Line 102-103, the author should provide the actual picture illustrating how to create different shade proportions.

Answer: We have incorporated the actual pictures from the field showing each shading condition in the revised version of Figure 1A.

-Line 104, how exactly was the density of the woven plastic nets?  The manufacture also need be provided.

Answer: We have included the information of density and manufacture in Material and Methods part (line 106-108).

-Line 165-170, how many volume of essential oil was dipped on the plates?

Answer: We performed the in vitro experiment with final concentration of essential oil 4 µg/mL.

-There is a lack of data on essential oil production under different shade conditions, such as yield.

Answer: We have included the yield (v/w) of essential oil from each shade as presented in Figure 1B.

This manuscript is a resubmission of an earlier submission. The following is a list of the peer review reports and author responses from that submission.

Round 1

Reviewer 1 Report

Comments and Suggestions for Authors

The manuscript investigates the chemical composition and bioactivity of Curcuma xanthorrhiza essential oil extracted from distinct shaded plants. Overall, the work is well-designed and well-executed. The abstract and introduction clearly state the purpose of the study. After carefully reading this work, I have some observations:

1.      I suggest rereading the text for typos and mistakes.

2.      Lines 84-86: relocate to the discussion.

3.      Line 97: How were the plant rhizomes obtained? From where? In what condition? Who identified the plant material?

4.      Line 108: Add the sample prep process. What were the storage conditions? How long were the samples stored before analysis? Were the samples washed? Were the rhizomes chopped, sliced, or ground? Clarify in the text.

5.       Line 109: References appear in two styles. Please follow the journal’s instructions for citations.

6.      Line 162: Avoid starting a sentence with a number not written out.

7.      Figure 1a is not clear.

8.      Lines 193 and 218: avoid using references in the results section. This section should focus on describing the findings of the current study. Rewrite or relocate these parts.

9.      Table 1: The authors mention 3 replicates of each treatment in the methods. Add statistics.

Comments on the Quality of English Language

The work would benefit from close editing.

Author Response

We would like to thank reviewers for valuable feedback. We have revised our manuscript accordingly to reviewers’ suggestions. Our responses to each point addressed by reviewers are written below.

Reviewer 1

The manuscript investigates the chemical composition and bioactivity of Curcuma xanthorrhiza essential oil extracted from distinct shaded plants. Overall, the work is well-designed and well-executed. The abstract and introduction clearly state the purpose of the study. After carefully reading this work, I have some observations:

  1. I suggest rereading the text for typos and mistakes.

Answer: Thank you for the suggestion. We have proofread our manuscript and corrected the typos and mistakes in the revised version.

  1. Lines 84-86: relocate to the discussion.

Answer: Thank you for the suggestion. We have relocated this sentence to Conclusion.

  1. Line 97: How were the plant rhizomes obtained? From where? In what condition? Who identified the plant material?

Answer: We have addressed this question in the methods section 2.2, which is now updated in the revised version of the manuscript. We planted the plants in Tropical Biopharmaca Cultivation Conservation, Unit Park of IPB University. The identification of plant material was performed by an expert in Tropical Biopharmaca Research Center.

  1. Line 108: Add the sample prep process. What were the storage conditions? How long were the samples stored before analysis? Were the samples washed? Were the rhizomes chopped, sliced, or ground? Clarify in the text. (harvest, wash, directly to essential oil extraction by distillation, slice 5 mm).

Answer: We have addressed this question in the methods section 2.2. More detailed information has been provided in the revised version of the manuscript.

  1. Line 109: References appear in two styles. Please follow the journal’s instructions for citations.

Answer: Thank you for the valuable comment. We have revised our reference section following the Journal’s instructions.

  1. Line 162: Avoid starting a sentence with a number not written out.

Answer: Thank you for the suggestion. We have revised this part accordingly.

  1. Figure 1a is not clear.

Answer: Thank you for the suggestion. We have revised Figure 1a and more detailed information has been added.

  1. Lines 193 and 218: avoid using references in the results section. This section should focus on describing the findings of the current study. Rewrite or relocate these parts.

 Answer: Thank you for the suggestion. We have revised this part accordingly.

  1. Table 1: The authors mention 3 replicates of each treatment in the methods. Add statistics.

Answer: Thank you for the valuable comment. In this experiment, we pooled the 3 replicates of the extract from each shade and performed the chemical composition analysis. We have added this information in method section 2.3 (line 115-116).

Reviewer 2 Report

Comments and Suggestions for Authors

The article presented by Waras Nurcholis and co-authors deals with the important plant Curcuma xanthorrhiza as a source of oils. Overall the article is well written but some issues need to be clarified and added:

Experiments and results

The authors should clarify what the cell line MCF-7 is.

The authors study the effect of C. xanthorrhiza EO on cancer cells but where is the control with non-cancer cells?  The authors should study the effect of C. xanthorrhiza EO on typical human cells to see if EOs are toxic.

In addition, the authors should show how the morphology of plants grown in the shade changed (did it affect plant biomass) compared to plants in full light.

Discussion

I suggest working on the discussion. The authors compare C. xanthorrhiza with species such as Rosmarinus officinalis, Mentha piperita, Thymus vulgaris, Origanum vulgare, and Myrtus communis however, is such a comparison justified? These plants produce aromatic substances in other parts and in a different way than C. xanthorrhiza.  In addition, C. xanthorrhiza accumulates spare materials in the rhizomes so plants even with reduced photosynthesis (through shade) can use these reserves to synthesize organic compounds.  This should be properly included in the discussion.

Round 2

Reviewer 1 Report

Comments and Suggestions for Authors

The authors have addressed most of the comments and suggestions. Minor suggestions were missed, such as adding statistics (SE or SD) to Table 1 and improving fig 1A. 

Author Response

Thank you for the comments! In case of Table 1, descriptive statistics cannot be added because the samples were pooled together before subjecting them to gas chromatography-mass spectrometry (GC-MS) analysis. This is now indicated in the legend of Figure 1 and in 2.3. of the revised manuscript. A new, improved version of Figure 1 was included into the revised version as well.

Reviewer 2 Report

Comments and Suggestions for Authors

I asked that the authors perform a cytotoxicity test on human (non-cancerous) cells. Unfortunately, the authors did a school experiment on shrimp. This approach raises my doubts about the reliability of the entire work.

Author Response

Thank you for the insightful comment! Unfortunately, currently we do not have the resources to perform the requested additional experiment. Majority of the compounds with antiproliferative or cytotoxic effects, even applied in clinical settings, are not completely specific for cancer cells. In this study, we only intended to report that the shading of Curcuma xanthorrhiza affects the composition of compounds in the extracted essential oils. This can result in differences in their biological activity, e.g. can affect their antiproliferative effect on HeLa cells. Concluding that the investigated samples have anticancer effects is not supported by our presented data, since we did not investigate any cancer-specific pathways. Therefore we removed all statement with respect to the anticancer activity and we only state that the shading affected the antiproliferative effect of the essential oil samples.

Round 3

Reviewer 2 Report

Comments and Suggestions for Authors

The article continues with the shrimp test

Supplementary Materials: The following are available online at www.mdpi.com/xxx/s1, Figure S1: 386 The toxicity test of C. xanthorrhiza EO extracted from each shade was performed by brine shrimp 387 lethality test.

If the authors want to present a potential anti-tumor agent then they need to present the full methodology including tests on mammalian cells to show that the agent is not toxigenic to healthy cells. Without this, for me, the article makes no sense.

Author Response

Reviewer’s comments:

Supplementary Materials: The following are available online at www.mdpi.com/xxx/s1, Figure S1: 386 The toxicity test of C. xanthorrhiza EO extracted from each shade was performed by brine shrimp 387 lethality test.

If the authors want to present a potential anti-tumor agent, then they need to present the full methodology including tests on mammalian cells to show that the agent is not toxigenic to healthy cells. Without this, for me, the article makes no sense.

Response:

Following the suggestion from the reviewer, the result of the brine shrimp test is omitted. In the literature, it has been shown that curcuma extract has antiproliferative effects [https://doi.org/10.1016/j.bjp.2017.11.001; doi.org/10.13057/biodiv/d20102]. Our aim was only to compare the biological effects of essential oil extract produced in different shading conditions. The previously reported studies have provided evidence that the extract and essential oil from C. xanthorrhiza do not exhibit toxic effects on normal cells.